# Probing the Effects of Retinoblastoma Binding Protein 6 (RBBP6) Knockdown on the Sensitivity of Cisplatin in Cervical Cancer Cells

**DOI:** 10.3390/cells13080700

**Published:** 2024-04-18

**Authors:** Harshini Mehta, Melvin Anyasi Ambele, Ntlotlang Mokgautsi, Pontsho Moela

**Affiliations:** 1Division of Genetics, Department of Biochemistry, Genetics and Microbiology, Faculty of Natural and Agricultural Sciences, University of Pretoria, Pretoria 0002, South Africa; u16025483@tuks.co.za (H.M.); u23040743@tuks.co.za (N.M.); 2Institute for Cellular and Molecular Medicine, Department of Immunology and SAMRC Extramural Unit for Stem Cell Research and Therapy, Faculty of Health Sciences, University of Pretoria, Pretoria 0001, South Africa; melvin.ambele@up.ac.za; 3Department of Oral and Maxillofacial Pathology, School of Dentistry, Faculty of Health Sciences, University of Pretoria, Pretoria 0001, South Africa

**Keywords:** cervical cancer, retinoblastoma binding protein 6 (RBBP6), *p53*, *Bcl-2*, cisplatin, personalized medicine, apoptosis

## Abstract

Cervical cancer is a major cause of death in women despite the advancement of current treatment modalities. The conventional therapeutic agent, cisplatin (CCDP), is the standard treatment for CC; however, resistance often develops due to the cancer’s heterogeneity. Therefore, a detailed elucidation of the specific molecular mechanisms driving CC is crucial for the development of targeted therapeutic strategies. Retinoblastoma binding protein 6 (*RBBP6*) is a potential biomarker associated with cell proliferation and is upregulated in cervical cancer sites, exhibiting apoptosis and dysregulated *p53* expression. Furthermore, *RBBP6* has been demonstrated to sensitize cancer cells to radiation and certain chemotherapeutic agents by regulating the *Bcl-2* gene, thus suggesting a crosstalk among *RBBP6/p53*/*BCL-2* oncogenic signatures. The present study, therefore, investigated the relationship between cisplatin and *RBBP6* expression in CC cells. Herein, we first explored bioinformatics simulations and identified that the *RBBP6/p53/BCL-2* signaling pathway is overexpressed and correlated with CC. For further analysis, we explored the Genomics of Drug Sensitivity in Cancer (GDSC) and found that most of the CC cell lines are sensitive to CCDP. To validate these findings, *RBBP6* was silenced in HeLa and Vero cells using RNAi technology, followed by measurement of wild-type *p53* and *Bcl-2* at the mRNA level using qPCR. Cells co-treated with cisplatin and siRBBP6 were subsequently analyzed for apoptosis induction and real-time growth monitoring using flow cytometry and the xCELLigence system, respectively. Cancer cells in the co-treatment group showed a reduction in apoptosis compared to the cisplatin-treated group. Moreover, the real-time growth monitoring revealed a reduced growth rate in *RBBP6* knockdown cells treated with cisplatin. Although wild-type *p53* remained unchanged in the co-treatment group of cancer cells, *Bcl-2* was completely repressed, suggesting that *RBBP6* is necessary for sensitizing cervical cancer cells to cisplatin treatment by downregulating *Bcl-2*. The Vero cell population, which served as a non-cancerous control cell line in this study, remained viable following treatment with both siRBBP6 and cisplatin. Findings from this study suggest that *RBBP6* expression promotes cisplatin sensitivity in HeLa cells through *Bcl-2* downregulation. Knockdown of *RBBP6* limits apoptosis induction and delays cell growth inhibition in response to cisplatin. The knowledge obtained here has the potential to help improve cisplatin efficacy through personalized administration based on the expression profile of *RBBP6* among individual patients.

## 1. Introduction

Cervical cancer (CC) is one of the most prevalent and prominent gynecological malignancies [1] and ranks the third most contributor to female cancer-related fatalities globally, after breast and colorectal lung cancer [1,2,3]. A total of 13% of CC cases are diagnosed in the later stages of progression, with a 5-year survival rate of 18.5% for metastatic disease compared to a substantially higher rate of 96.5% for localized cases [4]. On the contrary, individuals with early-stage and locally advanced CC benefit from conventional therapeutics, including resection surgery and radio-chemotherapy [5]. Given the heterogeneity of CC, understanding its specific mechanisms is imperative for developing effective prevention strategies and precise treatment approaches. Chemotherapy is the standard therapeutic for individuals with recurring CC [6]. The chemotherapeutic drug known as cisplatin (CDDP) has been demonstrated to exhibit significant efficacy as a treatment for advanced CC [6,7]. CDDP is a platinum-based chemotherapeutic drug that was initially shown to inhibit cell division in *Escherichia coli* (*E. coli*) bacteria [8,9,10,11]. This metallic-based drug is widely used to treat advanced stages of cancer, including those of the lung, breast, brain, bladder, and several others [12,13,14]. However, the emergence of resistance to cisplatin in patients significantly compromises its effectiveness in the treatment of recurrent CC [15,16,17,18]. Cisplatin’s anticancer effect is intricately associated with the modulation of specific interlinked molecular signaling pathways; therefore, understanding the molecular mechanism involved in CC is necessary for improved therapeutic approaches to treat recurrent or metastatic disease. Resistance to chemotherapy in CC involves various mechanisms [19,20].

The *p53* gene, also known as tumor protein (*p53*), serves as a pivotal tumor suppressor gene responsible for encoding the *Tp53* protein. This protein among its various functions, triggers cell cycle arrest in reaction to DNA damage or prompts apoptosis if the DNA is irreparable. The involvement of wild-type tumor suppressor *p53* (wt*p53*) plays a notably significant role in maintaining DNA integrity against anticancer drugs [21]. Wild-type *p53* acts as a suppressor of drug resistance, whereas mutant *p53* (mut-*p53*) promotes resistance to chemotherapy drugs [19,22]. Moreover, *p53* functions as a transcription factor in regulating the expression of various genes responsible for preserving genomic integrity, including the initiation of cell cycle arrest and DNA repair, as well as promoting apoptosis in case of oncogenic stress [23,24]. Accumulating studies have demonstrated that *p53* is mutated in various cancers; however, most individuals with CC express wt*p53* [25,26]. Recent studies have reported that *p53* is negatively regulated by the retinoblastoma binding protein 6 (*RBBP6*) gene, a eukaryotic protein with a size of 250 kDa, responsible for embryonic development [27]. The *RBBP6* gene is alternatively spliced to generate three protein isoforms that are differentially expressed in different tumor stages, with more advanced cancers exhibiting higher levels of *RBBP6* compared to less advanced stages [28,29]. Other studies have shown that *RBBP6* promotes uncontrolled cell growth and is associated with apoptosis during carcinogenesis [30,31,32]. Also, more information is emerging regarding the role of *RBBP6* in cancer treatment; specifically, its potential to sensitize cancer cells to radiation and chemotherapeutic agents, such as camptothecin, staurosporine, and GABA [28,33]. Furthermore, the overexpression of *RBBP6* leads to cell cycle arrest, a common feature of tumorigenesis [34], and is strongly associated with tumor progression in cervical and esophageal cancer [35]. This suggests that *RBBP6* may serve a critical role in the malignant phenotype of human cancer [35,36,37]. Motadi et al. [38] have demonstrated that *RBBP6* mRNA and its protein products are expressed in human lung cancer. However, little is known regarding its clinical and pathological significance in various cancers, including CC. Due to its ability to interact with *p53* and facilitate its degradation, *RBBP6* is a promising candidate for targeted drug therapy in cancer treatment [39,40]. Moreover, since *RBBP6* is implicated in the degradation of *p53* within cells, there is a necessity to gain a detailed understanding of how it interacts with *p53* at the molecular level. CC progression involves suppressing apoptosis through inactivated tumor suppressor genes, such as *P53*, activated or mutated oncogenes, and overexpressed anti-apoptosis genes [41]. *BCL-2* (B-cell lymphoma/leukemia-2 gene) is one of the oncogenes and has been shown to modulate apoptosis when dysregulated in CC tissues [42,43], suggesting possible crosstalk among *RBBP6/P53*/*BCL-2* oncogenes in promoting CC progression, resistance to treatment, and metastasis. CDDP is already accessible to patients, and understanding its relationship with *RBBP6* can help improve its efficacy through personalized administration based on the expression profile of *RBBP6* in CC cases that exhibit resistance to treatment. In this study, we elucidated the effects of *RBBP6* on the chemosensitivity of CC cells to cisplatin specifically through the *Bcl-2* and *p53*-dependent pathways. Findings from this study provided new knowledge on the role of *RBBP6* in regulating *Bcl-2* and *p5* to sensitize CC cells to cisplatin.

## 2. Materials and Methods

### 2.1. Bioinformatics Analysis

We analyzed the interaction among *RBBP6/P53/BCL-2* proteins using the STRING database under a high confidence of 0.778, and a protein enrichment of *p =* 0.000227 was achieved. Moreover, a functional enrichment analysis, including biological process and biological processes/Kyoto Encyclopedia of Genes and Genomes (KEGG), were analyzed using the DAVID annotation tool (https://david.ncifcrf.gov/, accessed on 20 Decemeber 2023). We further utilized functional enrichment and interaction network analysis tool (FunRich) to visualize the analysis. To further analyze the dysregulation of *RBBP6/P53/BCL-2* oncogenes in cervical cancer, we used an independent tool, the Human Protein Atlas (HPA) (https://www.proteinatlas.org/, accessed on 21 Decemeber 2023), with *p* < 0.05 considered statistically significant. Moreover, we evaluated the response of cervical squamous cell carcinoma (CESC) cell lines when treated with cisplatin (CDDP) using the Genomics of Drug Sensitivity in Cancer (GDSC) dataset from the Sanger Institute (https://www.sanger.ac.uk/, accessed on 21 Decemeber 2023). To identify expression levels of the *RBBP6/P53/BCL-2* oncogenes in different cervical cancer cell lines, we explored the expression database web tool, https://www.ebi.ac.uk/gxa/21, accessed on December 2023, using the RNA sequence data of 789 commonly used human cancer cell lines.

### 2.2. In Vitro Analysis

#### 2.2.1. Materials

Normal and cervical cancer cell lines (HeLa and Vero) were purchased from the National Institute of Biomedical Innovation, Health and Nutrition (NIBIOHN), in Ibaraki City, Japan. Knockdown of *RBBP6* was achieved using Ambion’s Silencer^®^Select Pre-designed siRNAs supplied by LifeTechnologies^TM^, and cisplatin (CDDP), European Pharmacopoeia (EP) reference standard, was purchased from Sigma Aldrich (St. Louis, MO, USA).

#### 2.2.2. Cell Culture

HeLa and Vero cells were maintained in a Dulbecco’s Modified Eagle’s Medium (DMEM, catalog #61965-026, Kwartsweg 2, Bleiswijk, The Netherlands) supplemented with 10% (*v*/*v*) Fetal Bovine Serum (FBS, catalog #10493-106, 3 fountain drive paisley pa4 9rf uk), 1% (*v*/*v*) Penicillin/Streptomycin antibiotics catalog #15140-122, 3175 Staley RdGrand Island, NY, USA) and 1% (*v*/*v*) fungizone, 3175 Staley Rd, Grand Island, NY, USA. The cells were incubated in a 37 °C humidified chamber supplied with 5% CO_2_. Cells were re-nourished with fresh media every 2 days after discarding old media and washing cells with a Phosphate-Buffered Saline (PBS, catalog #70011-036, 3 fountain drive paisley pa4 9rf uk) solution.

#### 2.2.3. Cell Viability Assay

An MTT (Methyl–Thiazolyl–Tetrazolium, catalog #M6494, 29851 Willow Creek RdEugene, OR, USA), assay was performed to identify the concentration of CDDP that inhibits 50% of cell growth (IC_50_) in HeLa cells. Monolayer cells were subcultured at ~70–80% confluency and seeded into 96-well plates (1 × 10^5^ cells/well) and then incubated overnight. The attached cells were subsequently treated with various concentrations of CDDP (50, 25, 12.5, 6.25, and 3.125 µg/mL), prepared by the serial dilution of 1 mg/mL stock solution in dH_2_O and incubation for 48 h. An MTT solution (5 mg/mL) was added to each well and incubated for 4 h. The resulting formazan crystals were resuspended in dimethyl sulfoxide (DMSO), and the absorbance was measured at 570 nm using a SpectraMax^®^ Paradigm^®^ multi-Mode microplate reader (Thermo Fisher Scientific, Waltham, MA, USA). Percentage cell viability was then calculated using the following formula:$$\%\;Cell\;Viability=\left[\frac{\left(Treated\;Absorbance-Blank\;Absorbance\right)}{\left(Untreated\;Absorbance-Blank\;Absorbance\right)}\right]\times 100$$

##### *RBBP6* Silencing and CDDP Treatment

HeLa and Vero monolayer cells were subcultured after they reached ~70–80% and resuspended in antibiotic-free media before seeding into 6-well plates. Cells were subsequently transfected using siRBBP6 obtained from the Ambion™ gene-specific Silencer^®^ (Life Technologies™, catalog #Am16708, Carlsbad, CA, USA), with sequences shown in Table 1, in complex with Lipofectamine™ 3000 (catalog #L3000-008, Thermo Fisher Scientific Inc.) lipid at a concentration of 100 pmol. Initially, the siRNA and transfection agent were diluted separately in Opti-MEM serum-free media (catalog #31985-047, Kwartsweg 2, Bleiswijk, The Netherlands) and then mixed in a 1:1 ratio. This was followed by incubation at room temperature for 15–20 min to form siRNA–lipid complexes before transfecting the cells. The cells were then incubated for 24 h to transiently silence *RBBP6*. Post-transfection cells were exposed to 25 µg/mL CDDP for an additional 24 and 48 h. The RNA was harvested for subsequent gene expression analysis.

#### 2.2.4. RNA Extraction

Total RNA was extracted from CDDP-treated HeLa and Vero cells post-transfection using TRIzol^®^ Reagent (catalog #15596018) following the manufacturer’s protocol. Cells were washed with cold PBS and resuspended in TRIzol^®^ Reagent for 5 min to dissociate nucleoprotein complexes. Cell debris was pelleted and following that, RNase-free chloroform was mixed into the remaining supernatant. The mixture was then vortexed for 15 s and centrifuged at 12,000 rpm for 15 min at 4 °C in a Prism™ R refrigerated microcentrifuge (Labnet International, Inc., Edison, NJ, USA) to achieve phase separation. The upper aqueous layer containing RNA was precipitated in isopropyl alcohol and centrifuged to obtain an RNA pellet, which was washed with 70% ethanol. The pellet was air dried for 10 min and resuspended in nuclease-free water. The RNA was further quantified using a NanoDrop^®^ ND-1000 Spectrophotometer (Inqaba Biotechnical Industries (Pty) Ltd., 525 Justice Mahomed St., Muckleneuk, Pretoria, South Africa) at absorbance A260/A280 nm, where ratios greater than 1.8 were considered pure. The 18s and 20s rRNA bands were visualized on a 2% ethidium bromide-stained agarose gel to confirm RNA integrity.

#### 2.2.5. Reverse Transcription

Total RNA from CDDP-treated HeLa and Vero cells post-transfection was reverse transcribed into cDNA using the PrimeScript™ RT Master Mix kit (Takara Bio Inc., Shiga, Japan, catalog #RR037A) according to the manufacturer’s guidelines. A 20 µL reaction mixture containing 5× PrimeScript™ RT Master Mix (composed of PrimeScript™ RTase, RNase Inhibitor, Oligo dT Primer, Random 6 mers, dNTP Mixture, and Mg2^+^-containing reaction buffer) and RNase-free dH_2_O were used to prepare the RNA template. Reverse transcription was performed in a T100™ Thermal Cycler (Bio-Rad Laboratories Inc., Hercules, CA, USA). The acquired cDNA was quantified using the NanoDrop^®^ ND-1000 Spectrophotometer at absorbance A260/A230 nm, where ratios greater than 2 were considered pure.

#### 2.2.6. Real-Time qPCR

Gene expression analysis of CDDP-treated HeLa and Vero cells post-transfection was performed in a 10 µL reaction mixture containing Luminaris™ Color HiGreen qPCR Master Mix (composed of SYBR Green (catalog #01321878), MgCl_2_, Taq polymerase, and dNTPs), forward and reverse primers specific to *RBBP6*, *P53*, and *BCL-2*, a 1000 µg/mL cDNA template, and nuclease-free dH_2_O. The housekeeping gene, *GAPDH*, was used as a reference to evaluate the quality of cDNA amplification. Primer sequences for target genes purchased from (Inqaba^®^ Biotechnical Industries (Pty) Ltd., Pretoria, South Africa) are shown in Table 2. Furthermore, RT-qPCR was performed in a CFX96™ Real Time System C1000™ Thermal Cycler (Bio-Rad Laboratories Inc., Hercules, CA, USA) under the following conditions: initial denaturation at 95 °C for 10 min followed by 45 cycles of denaturation at 95 °C for 15 s, primer annealing at 60 °C for 30 s, and extension at 72 °C for 30 s. Melt curve analysis was performed under the following conditions: 95 °C for 30 s followed by cooling to 65 °C for 5 s before the temperature was raised to 95 °C again at a rate of 0.5 °C/s with continuous fluorescence acquisition. Finally, the comparative CT Method (∆∆CT Method) was used to analyze the RT-qPCR results.

#### 2.2.7. Western Blot Analysis

HeLa and Vero cell lines, including a control group, were subjected to trypsinization for cell collection, subsequent to treatment with siRBBP6 and cisplatin (CDDP). Total protein extracts from both treated and untreated cells were obtained using a RIPA buffer (Thermofisher, 3747 N. Merdian Rd. RockFord, IL61101. Waltham, MA, USA). Following this, 20 µg of the protein extracts were separated using SDS-PAGE (Bio-Rad laboratoty Inc., Pretoria, South Africa), and were transferred onto polyvinylidene difluoride membranes using the transfer tank system, also from Bio-Rad. Subsequently, membranes were incubated with primary antibodies and left overnight in a −4 °C refrigerator; this was followed by a 1 h incubation with secondary antibodies the next day, and the specific antibodies used are listed in Table 3. Enhanced chemiluminescence (ECL) detection kits from Amersham Life Science, California city, CA, USA, were used to identify the proteins of interest, and image capture and analysis were performed using the ChemidocTM MP imaging system from Bio-Rad.

#### 2.2.8. Flow Cytometry

Apoptosis induction was measured in CDDP-treated HeLa and Vero cells post-transfection using the BD Pharmingen™ Annexin V-FITC Apoptosis Detection Kit I (BD Biosciences, San Jose, CA, USA, catalog #556547) following the manufacturer’s protocol. Cells were seeded in 6-well plates and transfected with siRBBP6 for 24 h. This was followed by treatment with apoptosis-inducing CDDP for an additional 24 and 48 h. The adherent cells were trypsinized, resuspended in growth media, and transferred into 15 mL tubes. The cell pellet was obtained by centrifuging at 1500 rpm for 5 min in an Eppendorf™ 5702 centrifuge. The pellet was subsequently washed with cold PBS before resuspending it in 100 µL of a 1× Binding Buffer at a concentration of 1 × 10^6^ cells/mL. In 1.5 mL tubes, the suspension cells were stained with 5 µL Annexin V-FITC and 5 µL Propidium Iodide (PI) and then incubated in the dark at room temperature for 15 min. Lastly, a 400 µL 1× Binding Buffer was added, and apoptosis was detected by flow cytometry within an hour in the CytoFLEX Flow Cytometer (Beckman Coulter Life Sciences, Brea, CA, USA). The data generated were analyzed using Kaluza Analysis software version 2.2.1 (Beckman Coulter Life Sciences, Brea, CA, USA).

#### 2.2.9. xCELLigence

Cell growth in CDDP-treated HeLa and Vero cells post-transfection was monitored using the xCELLigence System™ Real-Time Cellular Analysis (RTCA) instrument (Karl-Ferdinand-Braun-Straße, 228359 Bremen, Germany). Before cell seeding, background interference was taken into consideration. This is achieved by subjecting antibiotic-free media in 16-well E-plates to a current induced by the instrument. The 1.5 × 10^4^ cells/well cells were seeded into the 16-well E-plate and incubated for 24 h at 37 °C in the xCELLigence System™ instrument to monitor cell proliferation prior to further treatments. Once the logarithmic phase was reached, cells were transfected with siRBBP6 prior to exposing them to CDDP for an additional 24 and 48 h. Cell index (CI) values were recorded at 15 min interval sweeps for the duration of the experiment under the following conditions: 1st step: 1 sweep, 1 min interval, 00:00:39 total time; 2nd step: 100 sweeps, 15 min interval, 22:52:07 total time; 3rd step: 100 sweeps, 15 min interval, 47:38:30 total time; 4th step: 100 sweeps, 15 min interval, 72:12:25 total time.

#### 2.2.10. Statistical Analysis

The statistical significance of the differences observed for each series of experiments was determined using the paired Student’s *t*-test. Each experiment was performed using two technical and biological replicates. The results were expressed as the mean ± standard error of the mean (SEM) and presented as either *p* ≤ 0.05 (*), *p* ≤ 0.01 (**), or *p* ≤ 0.001 (***), where *p* ≤ 0.05 suggests that the differences between two groups are statistically significant and not due to chance.

## 3. Results

### 3.1. The Effects of RBBP6 Knockdown on p53 and Bcl-2 Gene Expression

To determine the co-expression among the *RBBP6/P53*/*BCL-2* oncogenic signature, we explored the protein–protein interaction (PPI) network analysis using STRING analysis. The interactions were examined by assessing correlations using experimental data (pink), gene neighborhoods (green), gene fusion (red), gene co-occurrences (blue), and gene co-expressions (black). Herein, we found that *RBBP6* with *TP53*, *RBBP6* with *MDM2*, *BCL2* with *RBBP6*, *BCL2* with *TP53*, *MDM2* with *TP53*, and *MDM2* with *BCL2* interacted within the same clustering network. The network had an initial three (3) proteins, which were further increased to thirty-four nodes. Protein enrichment of *p* = 0.000227 was obtained from the clustering analysis (Figure 1A). Following this, we utilized the DAVID database to conduct an in-depth analysis of enriched biological processes and KEGG pathways. Subsequently, FunRich software, version 3.1.4, was explored to construct the sets, with a stringent threshold set at *p* < 0.05. The identified affected biological processes specifically included those associated with cell communication, metabolism, and apoptosis, among others. The enriched KEGG or biological processes involved the regulation of retinoblastoma protein, the *P53* pathway, and ATM/ATR signaling pathways (responsible for regulating numerous signaling cascades, which respond to DNA strand breaking induced by deleterious agents) (Figure 1B,C). Prior to investigating the relationship between RBBP6 and CDDP sensitivity in HeLa and Vero cells, successful silencing of *RBBP6* was verified using RT-qPCR. The relative expression ratio of *RBBP6* reduced significantly to 0.22 ± 0.02 SEM (*p* < 0.05) in HeLa cells, indicating an approximate 78% reduction in expression 24 h post-transfection (Figure 1D). The relative expression ratio of *Bcl-2* in *RBBP6* knockdown HeLa cells reduced significantly to 0.54 ± 0.17 SEM (*p* < 0.05), indicating an approximate 46% decrease in expression post-transfection (Figure 1E). Although the relative ratio of *p53* increased to 1.29 ± 0.28 SEM, the increase in expression was not statistically significant (*p* > 0.05) relative to the untreated control (Figure 1F). In Vero cells, *RBBP6* was successfully knocked down by approximately 77% (Figure 1G) to a relative ratio of 0.23 ± 0.03 SEM (*p* < 0.05). Despite the reduction observed post-transfection, the changes in *Bcl-2* expression were not statistically significant relative to the untreated control (Figure 1H). *p53*, despite being absent in the untreated control, demonstrated an approximate 70% increase in expression after *RBBP6* was silenced (Figure 1I).

### 3.2. Analysis of the Sensitivity of RBBP6/P53/BCL-2 Oncogenes to Cisplatin Treatment

To analyze the sensitivity of *RBBP6/P53*/*BCL-2* oncogenes to cisplatin in cervical cancer, we explored an online bioinformatics database, the genomics of drug sensitivity in cancer tool (GSCA). Herein, we evaluated the response of different cervical squamous cell carcinoma (CESC) cell lines obtained from the catalog of somatic mutations in the cancer (COSMIC) project identified with a specific COSMIC ID. These cell lines included SKG-IIIa, SW756, SiHa, OMC-1, Ca-Ski, HeLa, ME-180, SISO, MS751, C-33-A, HT-3, DoTc2-4510, CAL-39, and C-4-I (Figure 2A). Interestingly, most of the cell lines responded to cisplatin treatment with low half-maximal inhibitory concentration (IC_50_). They are ranked by sensitivity as shown in Figure 2B.

### 3.3. RBBP6, Bcl-2, and p53 Gene Expression in Response to CDDP Treatment

CC cell lines were shown to be sensitive to CDDP using in silico simulations, as demonstrated in Figure 2. We further investigated the effects of CDDP in HeLa cells in vitro in order to validate these findings. Firstly, we explored the RNA-sequence data of 675 commonly used human cancer cell lines from the expression database web tool and identified expression levels of the *RBBP6*, *TP53*, *BCL-2*, and *MDM2* oncogenes in different CESC cell lines, including HeLa cells (Figure 3A). Moreover, the IC_50_ of this drug was further determined using an MTT assay. Cell viability was expressed as a percentage relative to an untreated control (100% viable). CDDP at 50 µg/mL and 25 µg/mL reduced cell viability significantly to ~30% ± 4.1% (*p* < 0.05) and ~45% ± 2.1% (*p* < 0.05), respectively. All concentrations below 25 µg/mL (12.5, 6.25, and 3.125 µg/mL) had no significant cytotoxic effect in these cells relative to the untreated control, as observed by the unexpected increase in percentage viability at these concentrations. Therefore, the working inhibitory concentration of CDDP used in this study was 25 µg/mL (Figure 3B). Furthermore, the effects of CDDP treatment on *RBBP6*, *Bcl-2*, and *p53* mRNA expression in HeLa and Vero cells were examined after 24 and 48 h exposure periods. The relative expression ratio of *RBBP6* did not change significantly (*p* > 0.05) after 24 h of CDDP exposure in HeLa cells (Figure 3B). Interestingly, 48 h of exposure promoted a significant increase in the relative expression ratio of RBBP6 to 1.38 ± 0.06 SEM (*p* < 0.05), indicating an approximate 38% increase in expression (Figure 3C). Despite an observed decrease in the expression of *Bcl-2* after 24 and 48 h of CDDP treatment, the differences were non-significant (*p* > 0.05) relative to the untreated control (Figure 3D). As expected, significant changes in the *p53* expression level were observed after CDDP treatment, where it increased by ~3-fold (3.12 ± 0.26 SEM, *p* < 0.05) and ~ 8-fold (8.15 ± 0.88 SEM, *p* < 0.05) after 24 and 48 h, respectively (Figure 3E). In Vero cells, CDDP reduced the expression level of RBBP6 significantly to 0.21 ± 0.04 SEM (*p* < 0.05) and 0.07 ± 0.005 SEM (*p* < 0.05) after 24 and 48 h of exposure, respectively (Figure 3F). Following CDDP treatment, *Bcl-2* exhibited less than ~4% (*p* < 0.05) expression after both exposure intervals (Figure 3G). No *p53* expression was observed, suggesting the complete absence of this tumor suppressor gene in CDDP-treated cells. CDDP, therefore, promoted RBBP6 expression in HeLa cells while inhibiting it in Vero cells.

### 3.4. Gene Expression in Response to Combined RBBP6 Knockdown and CDDP Treatment

To validate the dysregulation of *RBBP6/P53*/*BCL-2* oncogenes in CC tissue compared to adjacent normal tissues, we queried the Human Protein Atlas (HPA) database and found that the protein expression levels of *RBBP6*, *P53*, and *BCL-2* were significantly higher in CC tissues compared to those in normal tissues. Accordingly, *RBBP6* displayed high staining, strong intensity, and high staining quantity > 75%. *TP53* showed high staining, strong intensity, and high staining quantity > 75%. *BCL-2* displayed medium staining, moderate intensity, and high staining quantity > 75% (Figure 4A–C). Additionally, GECO, a gene expression correlation analytical tool, demonstrated significant positive correlations: *RBBP6* with T*P53*, *RBBP6* with *BCL-2*, and *TP53* with *BCL-2*, as indicated by positive Pearson correlation coefficients and *p*-values < 0.05 (Figure 4D–F). Having observed the effects of *RBBP6* knockdown and CDDP treatment separately on gene expression, we were interested in investigating the combined effect of *RBBP6* knockdown and CDDP treatment on *RBBP6*, *Bcl-2*, and *p53* expression in HeLa and Vero cells after 24 and 48 h exposure periods. In HeLa cells, *RBBP6* expression significantly increased by 87% in the relative expression ratio (1.87 ± 0.17 SEM, *p* < 0.05) after 24 h of CDDP exposure post-transfection. However, no expression of *RBBP6* was detected after 48 h of CDDP exposure post-transfection (Figure 4G). A similar pattern was observed with *Bcl-2*, where 24 h of CDDP exposure had no significant effect (*p* > 0.05) on expression levels but 48 h of exposure completely hindered expression levels (Figure 4H). After 24 h of CDDP exposure post-transfection, *p53* expression significantly increased by approximately 6-fold (5.97 ± 0.38 SEM, *p* < 0.05). However, after 48 h of exposure, no significant difference (*p* > 0.05) in *p53* expression was observed relative to the untreated control (Figure 4I). In Vero cells, CDDP treatment of transfected cells reduced *RBBP6* expression to less than ~10% (*p* < 0.05) after both exposure intervals (Figure 4J), while *Bcl-2* expression reduced almost completely (*p* < 0.05) after co-treatment (Figure 4K). *p53* was completely repressed, as was observed by the lack of amplification of this gene in co-treated samples. Accordingly, Western blot analysis demonstrated enhanced synergistic effects of siRBBP6 and CDDP on *RBBP6* and *BCL-2* signatures; however, the combination treatment was able to ameliorate the expression levels of the *p53* oncogene in both HeLa cell lines. GAPDH was used as an internal control (Figure 4L).

### 3.5. Apoptosis Detection Assay

Apoptosis induction was analyzed by flow cytometry in cells following transfection and/or CDDP treatment for 24 and 48 h exposure periods. The cells are gated into four different categories based on differential Annexin V-FITC and PI staining: viable cells (Annexin V-FITC and PI negative), early apoptotic cells (Annexin V-FITC positive, PI negative), late apoptotic cells (Annexin V-FITC positive, PI positive), and necrotic cells (Annexin V-FITC negative, PI positive). Cells undergoing early and late apoptosis were identified relative to an untreated control. After exposing HeLa cells to CDDP for 24 h, 62.3% of the population remained viable, while 32.4% and 5.3% underwent apoptosis and necrosis, respectively. After 48 h of CDDP exposure, the percentage of apoptotic cells increased to 77%, while 13.6% underwent necrosis. Combined with *RBBP6* gene silencing, CDDP exposure after 24 h induced 23% apoptosis and 11.8% necrosis, while 65.3% remained viable for CDDP treatment for 48 h post-transfection, inducing 59.4% apoptosis and 40.3% necrosis (Figure 5A–F and accompanying Table 4). In Vero cells, knockdown of *RBBP6* and/or treatment with CDDP for 24 and 48 h exposure periods induced less than or equal to 5% apoptosis and necrosis across all treatments (Table 5 and Figure 6A–F).

### 3.6. Real-Time Cell Growth Analysis

Cell proliferation following transfection and/or CDDP treatment was monitored in HeLa and Vero cells over a period of 72 h using the xCELLigence RTCA system. Cell growth was measured through the real-time detection of changes in electrical impedance and the cell index (CI) in both HeLa (Figure 7) and Vero (Figure 8) cells. The growth-inhibiting effect of *RBBP6* knockdown was validated through a steady decline in CI from the 48th hour following transfection with siRBBP6, as shown by the green growth curves. The treatment of HeLa cells with CDDP at a concentration of 25 µg/mL resulted in higher cell growth inhibition compared to CDDP at 12.5 µg/mL. Specifically, the cells treated with the lower dose (12.5 µg/mL) of CDDP remained in a plateaued state over the exposure period (Figure 7, turquoise blue growth curve), whereas cells treated with a 25 µg/mL CDDP dose showed growth reduction for approximately 42 h before they started to recover (Figure 7, blue growth curve). Interestingly, the combination of siRBBP6 and CDDP (25 µg/mL) reduced HeLa cell growth significantly, surpassing the growth inhibition achieved by CDDP alone. Although the combination of siRBBP6 and CDDP (25 µg/mL) inhibited cell growth significantly, a lower CDDP concentration was less effective. In Vero cells (Figure 8), a similar pattern of growth inhibition was observed when cells were exposed to different CDDP concentrations; however, this drug appeared to be highly cytotoxic, as shown by the significant reduction in cell growth after treatment. On the contrary, the combination of siRBBP6 and CDDP (25 and 12.5 µg/mL) had a minimal effect on Vero cell growth inhibition compared to HeLa cells, as indicated by the similar growth pattern observed at both drug concentrations (Figure 8). Therefore, combined treatment proved to be less cytotoxic compared to CDDP treatment alone.

## 4. Discussion

*RBBP6* is a proliferation-associated protein that is highly expressed during embryonic development and in rapidly dividing testis-derived cells. The expression of *RBBP6* is also associated with pathogenesis, specifically malignancies, where it is widely reported to play a role in apoptosis and cell proliferation owing to its ability to bind to tumor suppressors *p53* and *pRB* [28,34]. The DWNN domain, which is evolutionarily conserved across all three isoforms of *RBBP6*, has an E3 ligase activity that allows *RBBP6* to regulate other proteins within the cell via the proteasomal pathway [44]. The expression of *RBBP6* in malignancies is also associated with the sensitivity of cancer cells to certain chemotherapeutic agents and radiotherapy. In the present study, we demonstrated the potential role of *RBBP6* expression on the sensitivity of cervical cancer (CC) cells to cisplatin chemotherapy. Our findings revealed that CC cells treated with cisplatin underwent a significant apoptosis induction coupled with wild-type *p53* upregulation. Similarly, *RBBP6* knockdown resulted in apoptosis induction and a significant downregulation of the *Bcl-2* anti-apoptotic gene in these cells. However, prolonged exposure of *RBBP6* knockdown CC cells to cisplatin delayed cell growth inhibition and resulted in a marked reduction in apoptosis. Therefore, the reduction in cisplatin-induced apoptosis in *RBBP6* knockdown cells suggests that the overexpression of *RBBP6* is necessary for the sensitivity of CC cells to cisplatin.

Cisplatin is an FDA-approved, platinum-based chemotherapeutic drug that is currently being used as a single agent and in adjuvant therapy for patients with advanced CC [45]. When administered as a single agent, CDDP results in a higher overall response rate in patients compared to other chemotherapeutic agents, thus proving to be a promising anticancer drug [45]. However, cancer cells develop resistance to this drug and prolong their own proliferation through the evasion of apoptosis, particularly through *Bcl-2* and *p53* -mediated pathways [18]. Cervical cancer cells demonstrated elevated levels of *Bcl-2* anti-apoptotic protein [46] as well as a reduced wild-type *p53* expression due to degradation by HPV E6-associated protein [47], a signature that promotes increased resistance to chemotherapy. Nonetheless, these cells illustrate an enhanced response to cisplatin, as shown by the significant apoptosis induction, a marked upregulation of wild-type *p53*, as well as a reduced *Bcl-2* expression. Our findings demonstrated that CC cells treated with CDDP display a significant spike in *p53* mRNA and a significant reduction in cell growth via apoptosis induction, which validates the proposed mechanism of CDDP-induced apoptosis in CC cells [47]. Interestingly, prolonged exposure to CDDP led to a significant increase in *RBBP6* expression in cancer cells, even in the presence of siRBBP6, suggesting a possible interaction between CDDP and *RBBP6*, as well as early insights into the possibility of *RBBP6* acting in alliance during cisplatin-induced apoptosis via the wild-type *p53* and *Bcl-2* expression signature.

*RBBP6* expression was knocked down successfully in human cervical cancer cells using RNA interference technology. This was followed by the evaluation of *Bcl-2* and *p53* gene expression in *RBBP6* knockdown cells, and as expected, *Bcl-2* mRNA expression reduced significantly. This is consistent with findings by Xiao et al. (2018), who showed that *RBBP6* promotes *Bcl-2* transcription in cancer cells, which consequently promotes continuous proliferation and evasion of apoptotic signals. In addition, wild-type *p53* mRNA expression levels increased in response to *RBBP6* gene knockdown, and although the increase was not statistically significant, this observation is coherent with previous studies where *p53* expression was upregulated in response to *RBBP6* knockdown [31,33,48]. *p53* is strongly believed to be regulated by *RBBP6* since it contains its binding domain; however, the exact mechanism is not yet clear. Previous studies suggest that *RBBP6* promotes ubiquitin-mediated degradation of the wild-type *p53* tumor suppressor protein using the E3 ligase activity within its DWNN domain [28,36,49]. These studies speculate that by either degrading *p53* or promoting its inhibitory interaction with *MDM2* [50], *RBBP6* reduces the expression of *p53* in cancer cells, thus promoting cell proliferation.

*RBBP6* expression was also reduced significantly by ~77% in green monkey kidney cells (Vero), which represent the non-cancerous control cell line in this study. As expected, a non-significant reduction in *Bcl-2* was observed post-transfection; however, a significant increase in *p53* mRNA expression was observed, which is most likely due to the cells’ DNA repair machinery responding to the temporary stress inflicted by transfection with foreign nucleotide material. This is further supported by the lack of apoptosis induction where almost 99% of the Vero cell population remained viable post-transfection, thus indicating that the upregulation of *p53* was temporary with no consequential apoptosis induction. Furthermore, real-time cell growth analysis revealed a ~2.2 decrease in CI, which is minor compared to the untreated control, further suggesting that the knockdown of *RBBP6* had no major impact on the growth of Vero cells or their sensitivity to CDDP.

The effects of *RBBP6* knockdown on CDDP sensitivity in HeLa cervical cancer cells were evaluated by measuring the mRNA expression levels of wild-type *p53* and *Bcl-2* post-cisplatin treatment. Although there was no change in *Bcl-2* expression after the exposure of *RBBP6* knockdown cells to cisplatin for 24 h, the cells showed an upregulation of wild-type *p53* that is consistent with the CDDP-only treatment. Also, prolonging the cisplatin exposure period to 48 h led to a complete repression of the *Bcl-2* gene in the knockdown cells. At the mRNA level, these results suggest that *RBBP6* silencing promotes the downregulation of *Bcl-2* in cisplatin-treated cells, thus leading us to anticipate an enhanced apoptosis induction. However, the level of cisplatin-induced apoptosis in *RBBP6* knockdown cells was reduced, suggesting that *Bcl-2* downregulation is not the only mechanism responsible for the observed apoptotic cell death. In the complete absence of *RBBP6* and *Bcl-2* expression, wild-type *p53* expression is expected to increase significantly to promote more apoptosis compared to when CDDP was administered alone. However, the lack of wild-type *p53* upregulation in response to CDDP post-transfection further suggests that *RBBP6* knockdown does not promote *p53*-mediated sensitization of the cells to cisplatin. These data, therefore, validate the relationship between *RBBP6* and *Bcl-2* expression as previously reported [33]; however, targeting *RBBP6* for knockdown does not increase CDDP-induced *p53* -mediated apoptosis. In fact, the reduction in apoptosis induction following co-treatment with siRBBP6 and cisplatin suggests that *RBBP6* is necessary for the sensitization of cervical cancer cells to cisplatin.

This is illustrated by the apoptosis detection data where 24 h of CDDP exposure post-transfection led to only ~23% of the cell population undergoing apoptosis compared to 33% in the CDDP-only treatment. A much higher increase in apoptosis induction was expected considering the significant upregulation of *p53* in these cells; however, the opposite took place, and a 10% reduction in apoptosis was observed. Furthermore, cells exposed to siRBBP6 and CDDP for 48 h underwent an 18% reduction in apoptosis, from 77% in the CDDP-only treatment to 59% in the combined treatment, with *RBBP6* and *Bcl-2* expression completely inhibited after 48 h of drug exposure. To further analyze, we performed Western blot analysis and demonstrated that *p53* expression increased significantly after the combination treatment of siRBBP6 and CDDP. This will thus promote apoptosis due to the absence of its supposed negative regulator. The compromised apoptosis induction in combined treatment despite wild-type *p53* upregulation at 24 h of exposure and *Bcl-2* repression at 48 h of exposure suggests that RBBP6 plays a role in the sensitization of CC cells to CDDP. This is also confirmed by the real-time monitoring of cell growth in the presence of siRBBP6 and CDDP treatment, which shows that knocking down RBBP6 delays cell death induced by CDDP. The large proportion of late apoptotic and necrotic cells in response to siRBBP6 and CDDP co-treatment can be attributed to a phenomenon called necroptosis, a mode of cell death that involves binding the TNF-α and Fas ligand to their respective cell surface receptors to activate a series of receptor-interacting protein kinase (RIPK)-related signal transduction events that promote the release of intracellular contents into the extracellular space [51]. This study stands among the pioneering investigations to document the role of *RBBP6* in the response of cervical cancer cells to CDDP. We, therefore, conclude that in the presence of *RBBP6*, the cancer cells would exhibit poor sensitivity towards CDDP, which correlates with previous studies that reported increased sensitivity of HPV E6 knockdown HeLa cells towards CDDP through the restoration of *p53* and the induction of apoptosis and senescence [47,52].

## 5. Conclusions

Findings in this study suggest that *RBBP6* expression promotes the sensitivity of HeLa cells to cisplatin through *Bcl-2* downregulation. Knockdown of *RBBP6* limits apoptosis induction and delays cell growth inhibition in response to cisplatin, desensitizing cells to this drug. The unexpected overexpression of *RBBP6* in cisplatin-treated cells provides early insights into a possible relationship between the two. Therefore, *RBBP6* serves as a promising biomarker for targeted therapy, and its expression increases the sensitivity of CC cells to new and existing chemotherapeutics. These discoveries illustrate the potential role of *RBBP6* in the sensitivity of CC cells to cisplatin chemotherapy. The data can be used in patient stratification based on the *RBBP6* expression profile during drug administration and, therefore, assist in decision making regarding personalized CC management. Future studies can include a more robust investigation that involves more CC cell lines, the measurement of *Bcl-2* and *p53* at the protein level, and overexpressing *RBBP6* in a cancer cell line that does not express the gene. Assessing the role of *RBBP6* in CC cell response to cisplatin in vivo using a cancer mouse model will potentially shed more light on the molecular mechanisms of cisplatin-acquired resistance.

## Figures and Tables

**Figure 1 cells-13-00700-f001:**
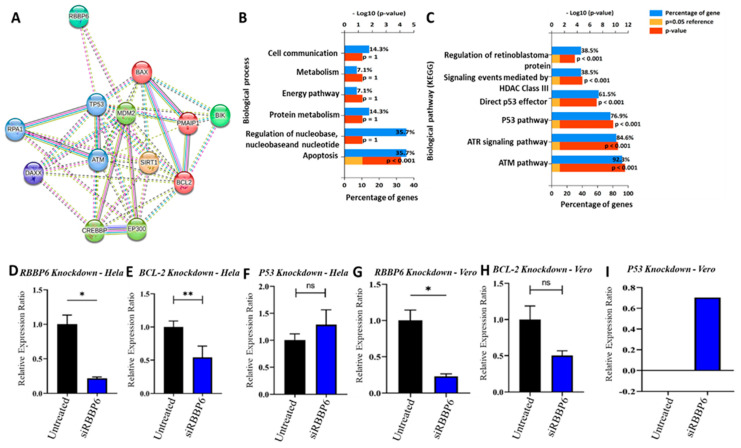
mRNA expression analysis in HeLa and Vero cells after transfection with 100 pmol siRNA targeting RBBP6. Experiments were performed in duplicates. (*) *p* < 0.05, (**) *p* < 0.01, (ns) *p* > 0.05. (**A**) protein–protein interaction of *RBBP6/P53*/*BCL-2* oncogenic signatures showing co-expression and co-occurrence within the same clustering network. (**B**) Affected biological process. (**C**) Enriched biological pathways when *RBBP6/P53*/*BCL-2* oncogenes are dysregulated in cervical cancer. Effect of *RBBP6* silencing on (**D**) *RBBP6*, (**E**) BCL-2, and (**F**) *p53* gene expression in HeLa cells. Effect of RBBP6 silencing on (**G**) *RBBP6*, (**H**) BCL-2, and (**I**) *p53* gene expression in Vero cells, with a *p*-value < 0.05 considered statistically significant.

**Figure 2 cells-13-00700-f002:**
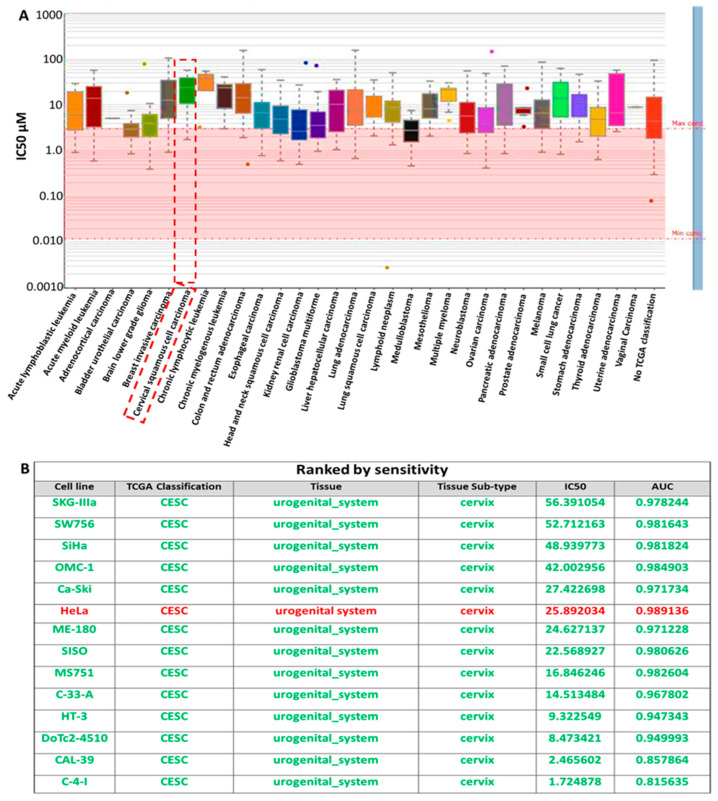
(**A**) Drug response of *RBBP6/P53*/*BCL-2* oncogenes to cisplatin (CDDP) in CESC cell lines. (**B**) The table shows the IC_50_ of CDDP on the treated cell lines.

**Figure 3 cells-13-00700-f003:**
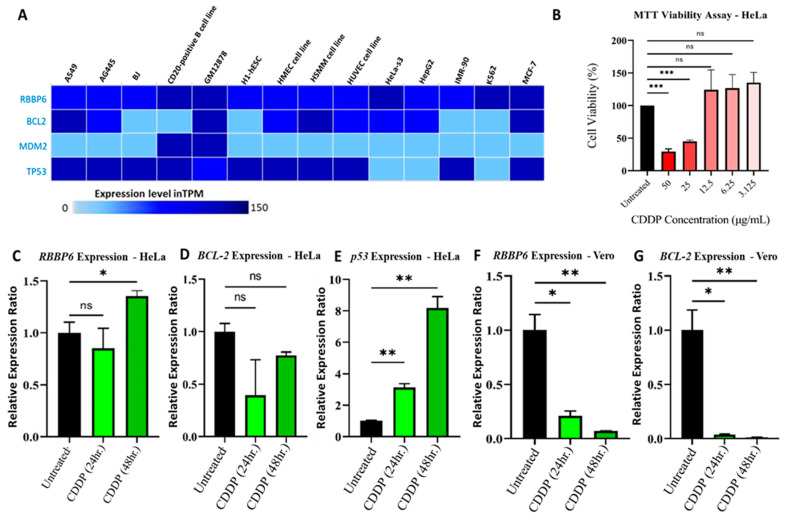
RBBP6, Bcl-2, and *p53* gene expression in response to CDDP treatment. (**A**) *RBBP6*, *TP53*, and *BCL-2* oncogenes are expressed in different CC cell lines, including HeLa cells. (**B**) Cell viability analysis using an MTT assay in HeLa cells treated with CDDP at 50, 25, 12.5, 6.25, and 3.125 µg/mL. Experiments were performed in triplicates. (***) *p* < 0.001, (ns) *p* > 0.05. mRNA expression analysis in HeLa CC cells after treatment with 25 µg/mL CDDP for 24 and 48 h exposure periods. Experiments were performed in duplicates. (*) *p* < 0.05, (**) *p* < 0.01, (ns) *p* > 0.05. Effect of CDDP treatment on (**C**) RBBP6, (**D**) BCL-2, and (**E**) *p53* gene expression in HeLa cells, as well as the effects of treatment in Vero kidney cells, as shown in (**F**) RBBP6 and (**G**) *BCL-2* gene expression. For the Vero cell lines, the experiments were performed in duplicates. (*) *p* < 0.05, (**) *p* < 0.01.

**Figure 4 cells-13-00700-f004:**
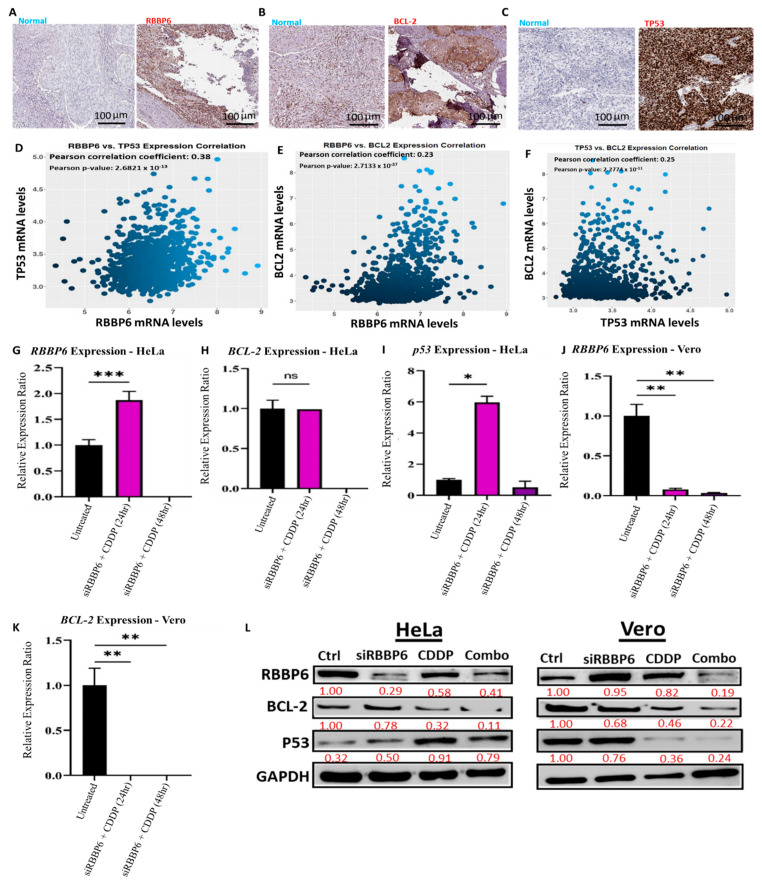
*RBBP6* knockdown and CDDP treatment synergistically reduced the expression of *RBBP6/P53*/*BCL-2* signatures in the CC (**A**–**C**) dysregulation of *RBBP6/P53*/*BCL-2* oncogenes in CC tissues compared to adjacent normal tissues acquired from the Human Protein Atlas (HPA). *RBBP6/P53*/*BCL-2* oncogenes positively correlate with each other in CC, which was demonstrated through positive Pearson correlation coefficients and statistically significant *p*-values below the threshold of 0.05. (**D**–**F**) mRNA expression analysis in HeLa CC cells after transfection with 100 pmol siRNA targeting *RBBP6* and co-treatment with 25 µg/mL CDDP for 24 and 48 h exposure periods. Experiments were performed in duplicates. (*) *p* < 0.05, (**) *p* < 0.01, (***) *p* < 0.001 (ns) *p* > 0.05. Effect of CDDP treatment post-transfection on (**G**) *RBBP6*, (**H**) *BCL-2*, and (**I**) *p53* gene expression in HeLa cells. mRNA expression analysis in Vero kidney cells after transfection with 100 pmol siRNA targeting *RBBP6* and co-treatment with 25 µg/mL CDDP for 24 and 48 hour exposure periods. Experiments were performed in duplicates. (**J**,**K**) The effect of CDDP treatment post-transfection on *RBBP6* and *BCL-2* gene expression in Vero cells. Western blot analysis showed significant synergistic effects of siRBBP6 with CDDP on *RBBP6*, *BCL-2*, and *P53* expression compared to CDDP alone. GAPDH was used as an internal control (**L**).

**Figure 5 cells-13-00700-f005:**
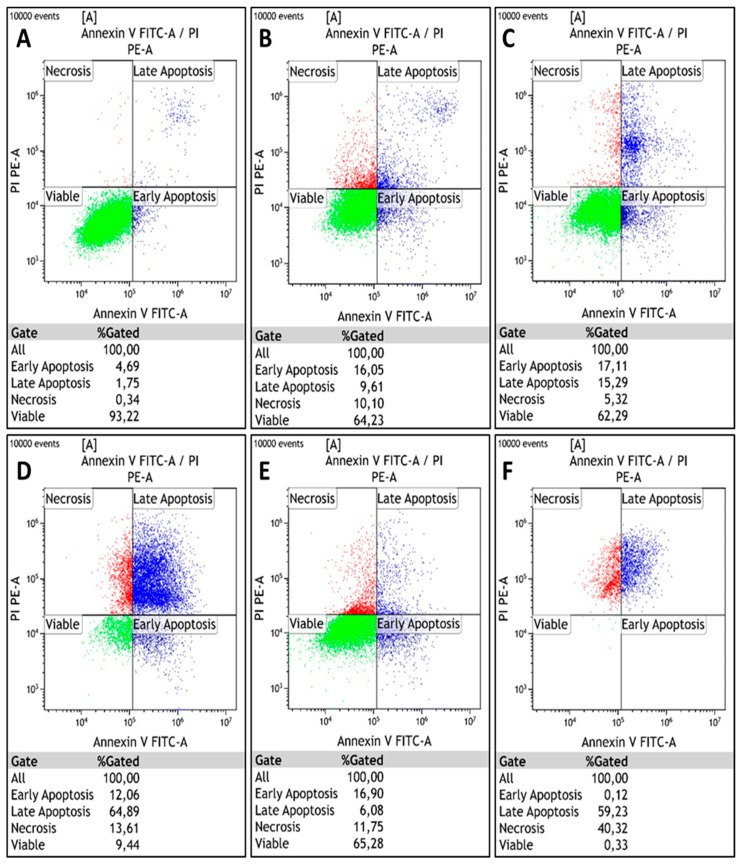
Flow cytometry analysis of apoptosis in HeLa cervical cells using Annexin V-FITC and PI following co-treatment with siRBBP6 and cisplatin for 24 and 48 h. Viable cells (green), early and late apoptotic cells (blue), and necrotic cells (red) were detected in (**A**) untreated cells and cells treated with (**B**) siRBBP6, (**C**) CDDP for 24 h, (**D**) CDDP for 48 h, (**E**) siRBBP6 and CDDP for 24 h, and (**F**) siRBBP6 and CDDP for 48 h. Table 4 shows the flow cytometry analysis of apoptosis in HeLa cells using Annexin V-FITC.

**Figure 6 cells-13-00700-f006:**
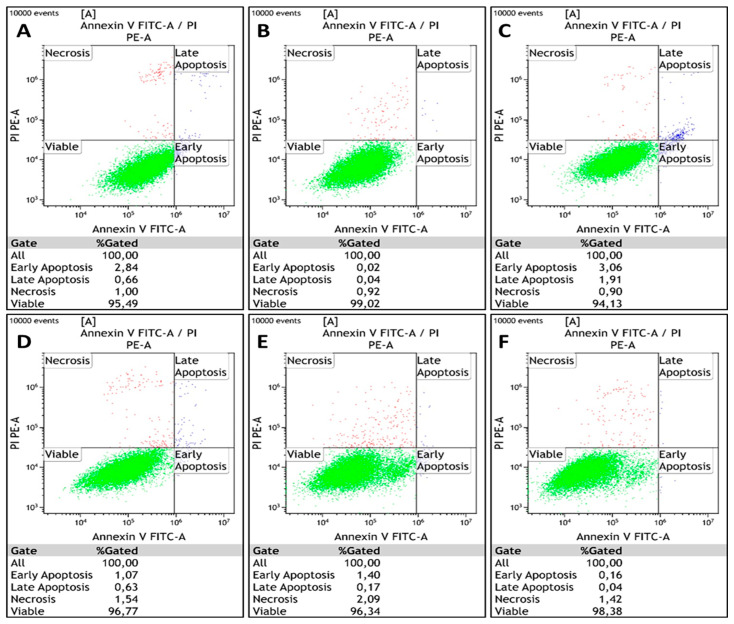
Flow cytometry analysis of apoptosis in Vero kidney cells using Annexin V-FITC and PI following co-treatment with siRBBP6 and cisplatin for 24 and 48 h. Viable cells (green), early and late apoptotic cells (blue), and necrotic cells (red) were detected in (**A**) untreated cells and cells treated with (**B**) siRBBP6, (**C**) CDDP for 24 h, (**D**) CDDP for 48 h, (**E**) siRBBP6 and CDDP for 24 h, and (**F**) siRBBP6 and CDDP for 48 h. Table 5 shows the flow cytometry analysis of apoptosis in Vero cells using Annexin V-FITC.

**Figure 7 cells-13-00700-f007:**
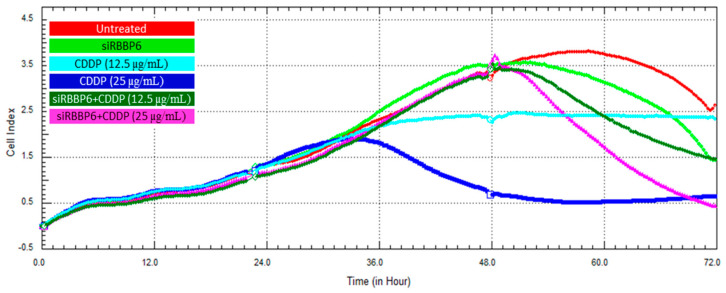
Cell growth analysis of HeLa cells using the xCELLigence RTCA system following RBBP6 knockdown and co-treatment with cisplatin over a period of 72 h. The proliferation of untreated cells (red), cells treated with siRBBP6 (green), 12.5 µg/mL CDDP (turquoise blue), 25 µg/mL CDDP (blue), and cells co-treated with siRBBP6 + 12.5 µg/mL CDDP (dark green) and siRBBP6 + 25 µg/mL CDDP (pink) was monitored in real time over a period of 72 h. Experiments were performed in duplicates.

**Figure 8 cells-13-00700-f008:**
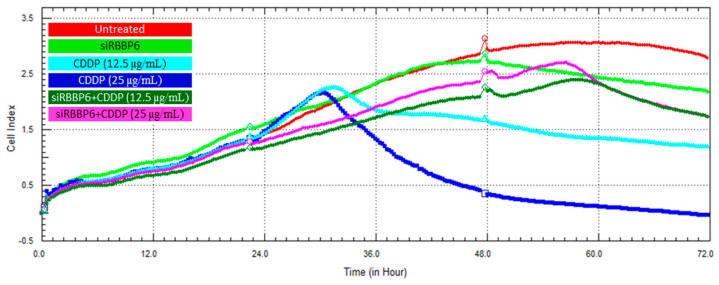
Cell growth analysis of Vero cells using the xCELLigence RTCA system following RBBP6 knockdown and co-treatment with cisplatin for 24 and 48 h. The proliferation of untreated cells (red), cells treated with siRBBP6 (green), 12.5 µg/mL CDDP (turquoise blue), 25 µg/mL CDDP (blue) only, and cells co-treated with siRBBP6 + 12.5 µg/mL CDDP (dark green) and siRBBP6 + 25 µg/mL CDDP (pink) was monitored in real time over a period of 72 h. Experiments were performed in duplicates.

**Table 1 cells-13-00700-t001:** RBBP6-specific siRNA sequence (Ambion™).

	Sense Strand	Antisense Strand
**Sequence**	5′-GCGAUGGCAACUACAAAAGtt-3′	5′-CUUUUGUAGUUGCCAUCGCtg-3′

**Table 2 cells-13-00700-t002:** Primer sequences for target genes.

	Forward Primer Sequence	Reverse Primer Sequence
** *RBBP6* **	5′-CAGCG ACGACTAAAAGAAGAG-3′	5′-GAGCGGCTGAATGATCGAGA-3′
** *p53* **	5′-GACGCTAGGATCTGACTGC-3′	5′-GACACGCTTCCCTGGATTG-3′
** *Bcl-2* **	5′-AGCCAGGAGAAATCAAACAGAC-3′	5′-GATGACTGAGTACCTGAACCG-3′
** *GAPDH* **	5′-CAGCCGCATCTTCTTTTGCG-3′	5′-TGGAATTTGCCATGGGTGGA-3′

**Table 3 cells-13-00700-t003:** List of the specific antibodies.

Target	Dilution	Company and Catalog No.	Predicted MW (kDa)
GAPDH	1:5000	ABCAM, GAPDH, Rabbit mAb, ab181602	36
RBBP6	1:1000	ABCAM, RBBP6 (63) Rabbit mAb, ab237514	250
BCL-2	1:1000	Cell Signaling, BCL-2, Rabbit mAb, #2872	26–28
*P53*	1:1000	Cell Signaling, *P53*, Rabbit mAb, #9282	53
2nd Antibodies	1:5000	Cell Signaling, Anti-Rabbit IgG HPR-Licked, #7074	

**Table 4 cells-13-00700-t004:** Flow cytometry analysis of apoptosis in HeLa cells using Annexin V-FITC and PI following co-treatment with 100 pmol siRBBP6 and 25 µg/mL cisplatin for 24 and 48 h exposure intervals.

	Untreated	siRBBP6	CDDP 24 h	CDDP 48 h	siRBBP6 + CDDP 24 h	siRBBP6 + CDDP 48 h
**Viable cells (%)**	93.2	64.2	62.3	9.4	65.3	0.3
**Apoptotic (early/late) cells (%)**	6.4	25.7	32.4	77.0	23.0	59.4
**Necrotic cells (%)**	0.3	10.1	5.3	13.6	11.8	40.3

**Table 5 cells-13-00700-t005:** Flow cytometry analysis of apoptosis in Vero cells using Annexin V-FITC and PI following co-treatment with 100 pmol siRBBP6 and 25 µg/mL cisplatin for 24 and 48 h exposure intervals.

	Untreated	siRBBP6	CDDP 24 h	CDDP 48 h	siRBBP6 + CDDP 24 h	siRBBP6 + CDDP 48 h
Viable cells (%)	95.5	99.0	94.1	96.8	96.3	98.4
Apoptotic (early/late) cells (%)	3.5	0.1	5.0	1.7	1.6	0.2
Necrotic cells (%)	1.0	0.9	0.9	1.5	2.1	1.4

## Data Availability

Dataset available on request from the authors.

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
