# Peer review of "Probing the Effects of Retinoblastoma Binding Protein 6 (RBBP6) Knockdown on the Sensitivity of Cisplatin in Cervical Cancer Cells"

_cells, 2024, doi:10.3390/cells13080700_

Round 1
Reviewer 1 Report
Comments and Suggestions for Authors
The authors presented a research article aimed at evaluating the effects of cisplatin treatment in cervical cancer due to cancer heterogeneity and the dysregulation of a specific molecular axis. The focus of the authors is on Retinoblastoma Binding Protein 6 (RBBP6) as a potential biomarker linked to cell proliferation and its overexpression in cervical cancer sites. The study explores the relationship between cisplatin and RBBP6 expression, using bioinformatics simulations and experimental approaches. Silencing RBBP6 in cancer cells reduced apoptosis and slowed growth in response to cisplatin, indicating RBBP6's role in sensitizing cervical cancer cells to cisplatin by downregulating Bcl-2. The findings suggest potential improvements in cisplatin efficacy through personalized treatment based on RBBP6 expression profiles in individual patients. Overall, the research idea is interesting, however, the manuscript cannot be accepted for publication unless major revisions will be addressed. Please see the comment below:
1) The entire manuscript needs extensive English editing performed by a English native speaker;
2) In the abstract section please use the full form for “GDSC”. Same comment for the method section;
3) In the following sentences you cited very old references or studies that are not strictly related to the content reported in the manuscript. In this part and throughout the manuscript, please provide updated and pertinent references. For this purpose, please see:
- https://doi.org/10.1016/j.cell.2023.02.038
- https://doi.org/10.2147/JEP.S267383
- https://doi.org/10.2478/raon-2019-0018
- https://doi.org/10.1016/j.ygyno.2019.04.013
4) Please check the grammar of the following sentence: “To further ana- 118 lyze the dysregulation of RBBP6/P53/BCL-2 oncogenes in cervical cancer on we used independent tool; human protein atlas (HPA) (https://www.proteinatlas.org/) P < 0.05 as statistically significant.”;
5) Please report the sequence of the siRNA used;
6) I suggest to move the primers used in the RT-qPCR experiments in a new separate table;
7) Check the grammar “we to evaluated” in the following sentence: “Herein, we to evaluated the response of different Cervical squamous cell carcinoma (CESC) cell lines obtained from the catalog of somatic mutations in cancer (COSMIC) project and identified with specific COSMIC ID.”;
8) On page 14, it is not clear if the image proposed is Figure 7 or Figure 8. Anyhow, a figure is lacking. Please check this critical issue. In addition, a color legend on the right side of the images is strongly suggested;
9) Western blot experiments aimed at evaluating the protein levels of RBBP6, p53 and Bcl-2 are lacking. This is a critical issue of the study. Please address this limitation;
10) Please consider to reduce the length of the Discussion section.
Comments on the Quality of English LanguageThe manuscript needs extensive English editing before publication.
Reviewer 2 Report
Comments and Suggestions for Authors
The manuscript reveals the role of Retinoblastoma Binding Protein 6 (RBBP6) in the sensitivity of cisplatin (CDDP) in cervical cancer cells. Transient knockdown of RBBP6 using siRNA in Vero and HeLa cells followed by the treatment if CDDP was used to evaluate cell viability via MTT assay, gene expression via RT-qPCR analysis, and apoptosis assay using flow cytometry. Further, xCELLigence real-time cellular analysis was performed to monitor cell proliferation post transfection. These assays revealed that downregulation of RBBP6 resulted in decreased cell growth. Overall, the manuscript is well-written. However, minor revisions are required to improve the overall quality of the manuscript.
In 2.2.1. Materials, provide the catalog numbers for the pre-designed siRNAs used in the manuscript.
Include the source and/or catalog numbers for growth medium, FBS, penicillin/streptomycin, fungizone, MTT, lipofectamine 3000, Opti-MEM medium, Trizol, primers, etc.
Which method was used for RT-qPCR analysis?
Reviewer 3 Report
Comments and Suggestions for Authors
The manuscript is focused on the analysis of the functional connections between RBBP6 protein and p53/BCL-2. The scientific approach is based mostly on the sensitivity measurement of silenced cells towards cisplatin. Although the experimental part was performed exclusively on two cell lines, the results are interesting.
Minor points:
1. The Abstract should be self-explanatory. Please add the description of the term GDSC.
2. Section 2.1 – What species was indicated during the analysis with STRING?
3. Fig 1 – please add the comparison of the relative expression of RBBP6 and BCL-2 in untreated cells (Hela VS Vero), to compare the differences between the cell lines regarding proteins of interest.
Comments on the Quality of English Language1. The spaces (or dots) in the text should be used consequently. As the example (from the Introduction section only): “(…) cases.[5].”, “(…) approaches[7].”, “(…) bacteria. [10-13].”, “(…) drugs [22].Wild-type”, “(…) CCcells”. Please check the text for more mistakes.
2. In most of the temperature “degrees”, the symbol is underlined.
Round 2
Reviewer 1 Report
Comments and Suggestions for Authors
Dear Authors,
thank you for your prompt reply. Some of my previous comments were properly addressed, however, some severe issues still affect the validity of your findings. Please address the following points:
Previous comment n. 3 - The manuscript was not updated nor the references. I apologize for my previous confusing suggestion. I kindly request you to update the information presented in the following sentences:
"On the contrary, individuals with early-stage and locally advanced CC, benefit from conventional therapeutics including; resection surgery and radio-chemotherapy [6]. Given the heterogeneity of CC, understanding its specific mechanisms are imperative for developing effective prevention strategies and precise treatment approaches[7]. Chemotherapy is the standard therapeutic for individuals with recurring CC[8] The chemotherapeutic drug known as cisplatin (CDDP) has been demonstrated to exhibit significant efficacy as treatment for advanced CC[9]. CDDP is a platinum-based chemotherapeutic drug that was initially shown to inhibit cell division in Escherichia coli (E. coli) bacteria. [10-13]."
For this purpose, please see:
- https://doi.org/10.1016/j.cell.2023.02.038
- https://doi.org/10.2147/JEP.S267383
- https://doi.org/10.2478/raon-2019-0018
- https://doi.org/10.1016/j.ygyno.2019.04.013
Previous comment n. 9 - "9) Western blot experiments aimed at evaluating the protein levels of RBBP6, p53 and Bcl-2 are lacking. This is a critical issue of the study. Please address this limitation;"
Please take your time to address this critical issue. Avoid also salami publication reporting the WB data in an additional paper. If necessary, you can request a deadline extension for these revisions.
Round 3
Reviewer 1 Report
Comments and Suggestions for Authors
All my comments were properly addressed.